# Cardiac Fibrosis and Fibroblasts

**DOI:** 10.3390/cells10071716

**Published:** 2021-07-06

**Authors:** Hitoshi Kurose

**Affiliations:** Graduate School of Pharmaceutical Sciences, Kyushu University, Fukuoka 812-8582, Japan; hikurose@phar.kyushu-u.ac.jp

**Keywords:** fibroblasts, myofibroblasts, fibrosis, single-cell analysis, genetic lineage tracing

## Abstract

Cardiac fibrosis is the excess deposition of extracellular matrix (ECM), such as collagen. Myofibroblasts are major players in the production of collagen, and are differentiated primarily from resident fibroblasts. Collagen can compensate for the dead cells produced by injury. The appropriate production of collagen is beneficial for preserving the structural integrity of the heart, and protects the heart from cardiac rupture. However, excessive deposition of collagen causes cardiac dysfunction. Recent studies have demonstrated that myofibroblasts can change their phenotypes. In addition, myofibroblasts are found to have functions other than ECM production. Myofibroblasts have macrophage-like functions, in which they engulf dead cells and secrete anti-inflammatory cytokines. Research into fibroblasts has been delayed due to the lack of selective markers for the identification of fibroblasts. In recent years, it has become possible to genetically label fibroblasts and perform sequencing at single-cell levels. Based on new technologies, the origins of fibroblasts and myofibroblasts, time-dependent changes in fibroblast states after injury, and fibroblast heterogeneity have been demonstrated. In this paper, recent advances in fibroblast and myofibroblast research are reviewed.

## 1. Introduction

Cardiac fibrosis is defined as a state in which excess extracellular deposition of collagens and extracellular matrix (ECM) occurs [1]. Cardiac fibrosis occurs when the heart is exposed to stresses, such as ischemic injury and chronic high blood pressure. Since fibrosis causes cardiac dysfunction, it is a target for treatment with drugs, medical devices, or tissue transplantation. ECM proteins, including collagen, are primarily produced by myofibroblasts, which are differentiated mainly from resident fibroblasts. Manipulation of the activity and number of myofibroblasts appears to be important for the inhibition of progression to more severe fibrotic states, and for recovery from the fibrotic state [2]. Since myofibroblasts play a central role in inflammation and fibrosis after cardiac injury [3], a detailed knowledge of fibroblasts and myofibroblasts is necessary when developing novel antifibrotic drugs. Recent technological advances have been used to address some of these issues. In this paper, the progress of research into fibroblasts, myofibroblasts, and fibrosis is presented.

## 2. Cardiac Cell Types

There are several types of cells in the healthy heart. Cardiomyocytes are cells that are responsible for contraction and relaxation. In addition to cardiomyocytes, the heart is made up of fibroblasts, endothelial cells, macrophages, mast cells, and lymphocytes, and a few other cell types. These cells interact with each other to maintain homeostasis in both healthy and diseased conditions [4]. In the resting state of a healthy heart, fibroblasts constantly modify the extracellular environment by producing and degrading the ECM [5]. The number of fibroblasts in the healthy heart is low. However, fibroblasts are differentiated to myofibroblasts when the heart is exposed to stresses, such as cardiac injury or chronic hypertension. Myofibroblasts actively produce collagen and other ECM proteins. Endothelial cells release autocrine and paracrine factors, such as nitric oxide and prostaglandins, which affect other types of cells [6]. Endothelial cells also express adhesion molecules that are involved in direct cell-to-cell interactions with myocytes or leukocytes. Macrophages residing in the healthy heart are different from macrophages infiltrating during cardiac injury. Resident macrophages play important roles in maintaining cardiac homeostasis, by facilitating atrioventricular conduction [7]. In injury, resident macrophages are replaced by monocyte-derived macrophages [8], which regulate fibrosis by secreting profibrotic cytokines and growth factors. Macrophages also play a role in matrix remodeling by secreting proteases. Mast cells are also present in the healthy heart. Mast cells expand in fibrotic areas and contribute to fibrosis, partly by the production of fibroblast-activating cytokines and growth factors [9,10,11]. Subsets of T lymphocytes may directly activate fibroblasts by producing profibrotic cytokines, interleukin (IL)-4 and IL-13. T lymphocytes may activate macrophages, inducing fibrosis. T lymphocytes may also affect the survival of cardiomyocytes, which promote the replacement of dead cells with fibrotic scarring [12]. Regulatory T cells (Tregs) are associated with the attenuation of myocardial fibrosis in various disease models of the heart [13], and myocardial infarction [14] through the modulation of macrophages [15] and fibroblasts [16]. Thus, each cell contributes differently to injury- or stress-induced cardiac fibrosis.

## 3. Classification of Fibroblasts

Among the multiple types of cells in the heart, fibroblasts are unique, since they differentiate into myofibroblasts and produce the ECM, which preserves the structural integrity of the injured heart. Recent histology-based and flow cytometric methods have demonstrated that fibroblasts account for about 13% of cells in the mouse heart [17,18]. When the heart is exposed to injury, such as myocardial infarction or hypertrophy, the fibroblasts differentiate into myofibroblasts, which produce ECM. Excess deposition of ECM causes fibrosis, leading to tissue dysfunction. The management of the number or function of myofibroblasts is important for the treatment of fibrosis.

Myofibroblasts have the following several important characteristics: extensive endoplasmic reticulum [19], the expression of contractile proteins, such as α-smooth muscle actin (α-SMA), and the synthesis of matricellular proteins, including periostin. Although α-SMA is widely used for the identification of myofibroblasts, not all α-SMA-expressing cells are myofibroblasts. Five days after the MI operation, α-SMA were expressed in subsets of PDGF receptor α- or collagen1α1-positive cells [20]. So far, specific and reliable marker proteins for myofibroblasts have not been reported.

The origins of myofibroblasts have been analyzed by labeling the various types of cells with reporter genes under cell-specific promoters, as part of lineage-tracing experiments [21]. Genetic labeling has advantages over the immunological detection of marker proteins, since maker proteins sometimes disappear during development or differentiation. Marker proteins can be expressed not only in cells that are of interest in a particular experiment, but also in functionally irrelevant cells. Lineage-tracing experiments allow the expression of reporter genes under the control of a cell type-specific promoter. This modification labels the cells permanently, even after the promoter activity turns off, and allows the identification and tracking of the cells once that promoter is activated.

Lineage-tracing experiments showed that fibroblasts that are present in the left ventricle and ventricular septum are derived from endocardial cells via the endothelial-mesenchymal transition (EndoMT), and epicardial cells through the epithelial-mesenchymal transition (EMT) [22]. A small number of fibroblasts are generated by the differentiation of neural crest cells. However, mature endothelial cells, epicardial cells, and bone marrow-derived cells do not contribute to the population of fibroblasts. After labeling the fibroblasts using cell type-specific promoter-reporter genes, the question of whether fibroblasts exhibit different functions depending on their origin was addressed. After pressure overload, epicardial-derived fibroblasts labeled with Tbx18 promoter-GFP, and endocardial-derived cells labeled with Tie2 promoter-GFP were isolated. The analysis of RNAs revealed similar expression profiles between epicardial-derived and endocardial-derived cells [23,24,25], and the two groups of cells had similar proliferative activity [26]. Therefore, it was concluded that there is no significant relationship between the origin of fibroblasts and their function.

Genetic lineage tracing was used to trace the fate of fibroblasts over time, after myocardial infarction [27]. The following four different states of fibroblasts were identified using the technique: resting fibroblasts, active fibroblasts, myofibroblasts, and matrifibrocytes. In one study, the Tcf21 promoter was used to label the fibroblasts in resting conditions, which correspond to tissue-resident fibroblasts. When the proliferative activity of fibroblasts was measured, using 5-ethynyl-2′-deoxyuridine (EdU) or immunodetection of Ki-67 after myocardial infarction, the fibroblasts with proliferating activity appeared two to four days after myocardial infarction (active fibroblasts), and the conversion of fibroblasts to myofibroblasts occurred four to seven days after myocardial infarction [27]. The myofibroblasts were derived from tissue-resident fibroblasts. These findings are consistent with those of another study [23]. Myofibroblasts were transformed to a new type of cell, called matrifibrocytes, 10 days after myocardial infarction [27]. Analysis of the expressed mRNAs suggested that cells from different states have distinct expression patterns. Fibroblasts in active states had high proliferative and migration activities. Myofibroblasts produced collagen and α-SMA. Matrifibrocytes are unique, since they localize at scar sites and express genes associated with bone and cartilage remodeling, including *Cilp2* and *Comp*. The physiological implications of these bone-related genes in the heart are not known.

The heterogeneity of fibroblasts has been demonstrated using single-cell RNA sequencing (scRNAseq), a relatively new, but rapidly developing, technology [28]. It allows the comprehensive characterization of gene expression and relationships in individual cells. Single-cell analysis of 11,492 cells revealed the heterogeneity of fibroblasts and cardiomyocytes during pressure overload-induced cardiac hypertrophy [29]. In this report, fibroblasts were grouped into six clusters, FB1 to FB6. FB1 corresponds to active fibroblasts in previous reports, and FB6 contains myofibroblast-like cells that express high levels of ECM and periostin. It is unknown whether the cells of each group contribute differently to fibrosis, and which groups correspond to the previous classification. In contrast to fibroblasts, cardiomyocytes were divided into four groups, FC1–FC4, based on their expressed proteins. The cells of each group expressed different combinations of proteins that are involved in muscle development, metabolism, and contraction. Among them, FC3 and FC4 are of interest, due to the expression of endothelial or fibroblast markers, such as cadherin 5, von Willebrand factor, vimentin, and decorin. FC3 and FC4 cells are not of fibroblast origin, since they did not express marker proteins that label fibroblasts, such as the transcription factor Tcf21 and the PDGFα receptor. Correlation analysis suggested that the changes in groups FC3 and FC4 are highly correlated to late-stage cardiomyocyte pathology. However, it remains to be determined whether FC3 and FC4 have specific functions in the progression of myocardial infarction-induced heart failure.

There are several studies that have used scRNAseq for the analysis of cellular states. Skelly et al. reported new cardiac fibroblast states in cells isolated from healthy hearts [30]. A new population of fibrocyte cells was identified, expressing markers of both fibroblasts and immune cells. However, the functional role of these cells in the heart, at baseline or injury, was not investigated. Farbehi et al. used lineage tracing to isolate the cells expressing PDGF receptor α, and sequenced them [31]. They identified novel myofibroblast subtypes, expressing both profibrotic and antifibrotic signatures. McLellan et al. studied the fibroblast populations that are present after angiotensin II infusion, using scRNAseq [32]. They did not detect myofibroblasts expressing αSMA. Instead, they identified two fibroblast subpopulations, expressing the matricellular proteins Cilp1, a mediator of cardiac ECM remodeling, and thrombospondin 4. Based on several scRNAseq studies, a new scheme of fibroblast states has been proposed.

In the scheme, there are four different states of fibroblasts (Figure 1) [33,34]. In the basal state, three subsets of fibroblasts were identified. After injury or during aging, fibroblasts activated by the stimulation of inflammatory cytokines acquire proliferating and ECM-producing activities, entering an expansion state. Fibroblasts in an expansion state are subdivided into three different subsets. The activated state of fibroblasts follows the expansion state. The function of fibroblasts in an activated state is almost equivalent to that of so-called myofibroblasts expressing α-SMA. The α-SMA-negative cells may be involved in the promotion of angiogenesis. Activated fibroblasts change their phenotypes in one of several ways (i.e., resolution). They may revert to resting fibroblasts, undergo apoptosis, or enter senescence. Some fibroblasts in an activated state can still produce ECM. Although multiple studies have demonstrated that fibroblasts can regulate inflammation, single-cell sequencing experiments have not identified fibroblasts with inflammatory activity. The marker proteins of each fibroblast state are shown in Table 1.

These results demonstrate the heterogeneity of fibroblasts, but the relationships and interactions between the cells in each state are unknown. Their functions and contribution to cardiac fibrosis remain to be determined in the future.

Although systemic sclerosis is a cause of cardiac fibrosis, many studies of cardiac fibrosis in animal models have been of myocardial infarction, pressure overload-induced hypertrophy, or treatment with neurotransmitters or hormones. It is unknown whether cardiac fibroblasts of systemic sclerosis exhibit states that are similar to those induced by myocardial infarction.

## 4. Differentiation of Fibroblasts Following Cardiac Injury

The possibility that fibroblasts convert to other cells, or vice versa, after maturation has been investigated [35]. The prolonged culture of macrophages resulted in cells that express various fibroblast markers, such as type I collagen, prolyl-4-hydroxylase, fibroblast-specific protein-1, and fibroblast activation protein (FAP) [35]. Animals that express yellow fluorescent protein (YFP) only in cells of myeloid lineage have been created. These marker fibroblast proteins were detected in infiltrating YFP-positive macrophages after myocardial infarction. Chlodronate liposome treatment to deplete macrophages reduced the number of collagen-positive fibroblast marker-expressing cells. These results suggest that fibroblasts are derived from macrophages. It is interesting to examine the contribution of the macrophage-fibroblast transition to cardiac fibrosis. Although the contribution of the macrophage-fibroblast transition to cardiac fibrosis is not unknown, inhibition of the transition to fibroblasts may help to reduce fibrosis after myocardial infarction.

It has been reported that endothelial cells are not a major source of fibroblasts in the adult mouse heart [24]. However, there is controversy about the conversion of endothelial cells to myofibroblasts. It has been reported that fibroblasts acquire endothelial cell-like phenotypes during ischemia-reperfusion [36,37]. A series of experiments, using mice with genetically labeled fibroblasts, demonstrated that 20%–40% of fibroblasts express various endothelial cell markers, and the isolated cells can form a capillary network. The expression of p53 was essential for the process of conversion from fibroblasts to endothelial cells [36]. The stimulation of p53 signaling improved cardiac dysfunction during ischemia-reperfusion. However, the opposite result was reported by a different group, who found that resident fibroblasts did not contribute to neovascularization after cardiac injury [37]. In the latter report, the pulse-chase labeling of fibroblasts after ischemia-reperfusion showed that resident fibroblasts did not express genes involved in angiogenesis, which are characteristic of endothelial cells. The origin of almost all endothelial cells was resident endothelial cells. Different approaches resulted in different conclusions. Thus, it may be necessary to confirm the findings using other techniques, such as scRNAseq and proteomic analysis of isolated cells.

## 5. Myofibroblasts as Phagocytes

Myofibroblasts have been recognized primarily as the cells that produce ECM components, such as collagen and fibronectin. They also interact with inflammatory cells through secreted factors. A new role for myofibroblasts in inflammation induced by myocardial infarction has been reported. Myofibroblasts, but not fibroblasts, efficiently phagocytose apoptotic cells and secret cytokines that suppress inflammatory responses [38]. This activity is similar to that of macrophages, which induce immuno-suppressive responses by the engulfment of apoptotic cells.

Myocardial infarction induces necrosis of cardiomyocytes. Phosphatidylserine is present on the surface of cells that have undergone necrosis, as observed at apoptosis. Therefore, terminal deoxynucleotidyl transferase-mediated dUTP nick end labeling (TUNEL) staining can be used as a marker of dead cells. Apoptotic cells are thought to be engulfed by phagocytes, such as macrophages. Nakaya et al. examined the expression of various molecules that are involved in engulfment after myocardial infarction [38] (Figure 2).

The expression of a factor called milk fat globule-EGF factor 8 (MFG-E8) was increased. MFG-E8 binds both phosphatidylserine expressed on the membranes of apoptotic cells and integrin present at the surface of phagocytic cells [39]. Since integrin does not directly bind phosphatidylserine, MFG-E8 functions as a bridge between apoptotic cells and phagocytes. MFG-E8 was found to be produced by myofibroblasts, and was used by myofibroblasts to phagocytose dead cells, releasing anti-inflammatory cytokines and preventing excessive inflammation. Myofibroblasts exhibit macrophage-like properties, such as the phagocytosis of dead cells and secretion of anti-inflammatory cytokines, thus behaving similarly to macrophages at myocardial infarction. Myofibroblasts are less active than macrophages in engulfing dead cells. However, fibroblasts easily differentiate to myofibroblasts at injury sites, and the number of myofibroblasts is believed to be high in ischemic areas during myocardial infarction. Thus, myofibroblasts compensate for their low ability to engulf cells by being present in high numbers. There is a major difference between myofibroblasts and macrophages. Unlike macrophages, myofibroblasts do not have antigen presentation activity [38]. Myofibroblast-mediated phagocytosis of apoptotic cells is an efficient way to prevent excess inflammation at an injured site. However, not all myofibroblasts phagocytose dead cells in vitro. Thus, distinct groups of myofibroblasts may have different functions, such as phagocytosis and differentiation to other types of cells.

It has been reported that cardiac fibroblasts contribute to inflammatory responses during the two to three days after myocardial infarction [40]. Fibroblasts are activated by damage-associated molecular patterns (DAMPs) released from dying or dead cardiomyocytes. Activated fibroblasts produce inflammatory cytokines and chemokines, as well as matrix-degrading proteins. These inflammatory cytokines activate fibroblasts to covert to myofibroblasts. Fibroblasts and myofibroblasts therefore form a positive loop to enhance the formation of myofibroblasts. Activated fibroblast- and myofibroblast-secreted chemokines recruit immune cells to an infarcted area. The recruitment of immune cells stimulates the clearance of matrix debris and promotes wound healing. Thus, myofibroblasts possess the following two different activities: phagocytosis-linked anti-inflammatory activity, and DAMP-induced inflammatory activity. The time courses of the two activities are different. Fibroblasts contribute to inflammation, to recruit immune cells during the early phase of inflammation, and promote the removal of dead cells together with immune cells, since myofibroblasts, but not fibroblasts, efficiently engulf dead cells. The removal of dead cells by immune cells and myofibroblasts then negatively regulates inflammation.

Experimental autoimmune myocarditis is a mouse model of CD4+ T-cell-mediated inflammatory cardiomyopathy. In experimental autoimmune myocarditis, heart-infiltrating CD133+ progenitor cells were reported to be converted to myofibroblast-like cells, with the help of TGF-β [41]. The type of disease may therefore determine the origin of myofibroblasts.

## 6. Signaling Controlling Differentiation to Myofibroblasts

TGF-β is a strong inducer of the differentiation of fibroblasts to myofibroblasts [42]. Inflammatory cells that are recruited to an injury site release cytokines, including TGF-β. Injury-associated cells also release alarmins and DAMPs [43]. These molecules cause inflammation, leading to the differentiation of fibroblasts to myofibroblasts. Thus, any inhibition of inflammatory responses will block the appearance of myofibroblasts, which eventually suppresses fibrosis. At the early stages of myocardial infarction, neutrophils are first recruited to the injury site [44,45]. Leukotriene B4 is a powerful attractant of neutrophils, which bind to the leukotriene B4 receptor BLT1. The inhibition of neutrophil recruitment by BLT1 gene knockout, or the use of a BLT1 blocker, decreased cardiac fibrosis by inhibiting inflammation [46,47].

Interleukins are important inflammatory cytokines. Multiple IL receptors are expressed in cardiac fibroblasts, which regulate fibroblast states and functions [44]. The effects of proinflammatory ILs on cardiac fibroblasts are blocked by cardiac fibroblast-specific deletion of IL receptors. Knockout of the IL11 receptor or IL17 receptor genes reduced injury-induced cardiac fibrosis and cardiac dysfunction [48,49]. These results reveal that cardiac fibroblast-specific deletion of IL receptor genes decreases the infiltration to, or activity of, immune cells in the area of the injury. They also suggest that cardiac fibroblasts play an important role in the regulation of injury-induced inflammation, through IL signaling.

After myocardial infarction, monocytes are mobilized from the bone marrow and differentiate into macrophages at the injury site [50]. Macrophages in mice can be depleted by treatment with chlodronate liposomes [51]. Macrophage-depleted mice showed decreased fibrosis and improved cardiac function. These results show that the inhibition of inflammatory signaling after myocardial infarction decreases fibrosis partly through the blockade of the differentiation into myofibroblasts.

TGF-β stimulation activates both the canonical Smad2/3 and the noncanonical mitogen-activated protein (MAP) kinase signaling pathways [42]. Fibrotic responses were inhibited by the knockout of fibroblast-specific TGF-β receptor 1/2, or the knockout of transcription factor Smad3, which is activated downstream of the TGF-β receptor [52]. TGF-β-Smad2/3 signaling of fibroblasts is a major factor in cardiac fibrosis induced by pressure overload. The knockout of TGF-β receptor 1/2 in fibroblasts also inhibited cardiac hypertrophy by pressure overload [52]. It appears that myofibroblasts interact with cardiac myocytes through direct cell-cell communication, or indirectly via a mediator that is secreted from myofibroblasts.

In addition to TGF-β, lysophosphatidic acid (LPA) stimulation induces the differentiation of fibroblasts to myofibroblasts [53,54]. LPA binds its own GPCRs to activate cellular responses [55]. In vitro, LPA stimulation induces fibrosis by the activation of the myocardin-related transcription factor-serum responsive factor (MRTF-SRF) pathway. Rho regulated actin oligomerization, which is regulated by the phosphorylation of actin by Rho kinase (ROCK). ROCK-mediated phosphorylation of actin increased the amount of the monomeric form of actin. Monomeric actin loses its activity to inhibit SRF. SRF, together with MRTF, activated the transcription of various genes, including profibrotic genes [56]. Compound CCG-203971 is a small-molecule inhibitor of the Rho-mediated MRTF-SRF pathway [57]. The administration of CCG-203971 inhibited bleomycin-induced lung fibrosis. Rho is a signaling molecule that is activated downstream of various receptors, including GPCRs and TGF-β receptors. Inhibitors of the Rho-ROCK pathway may suppress fibroblast activation and fibrosis in the heart more efficiently than receptor inhibition.

G-protein-coupled receptor kinase 2 (GRK2) is known to be a regulator of G-protein-coupled receptors (GPCRs), which act by phosphorylating agonist-bound GPCRs [58]. Cardiomyocyte-specific knockout of GRK2 demonstrated that GRK2 ablation protects the heart against cardiac dysfunction and fibrosis, following myocardial infarction [59]. These mice also showed a reduction in the development of heart failure following myocardial infarction. Fibroblast-specific knockout of GRK2, using the collagen 1α2 promoter, reduced the secretion of TNFα and suppressed the expression of profibrotic factors after ischemia-reperfusion [60]. The inhibition of fibroblast function improved cardiac dysfunction. These results show that the inhibition of GRK2 protects the heart against cardiac stresses in cardiomyocytes as well as in fibroblasts.

Transient receptor potential channel canonical 6 (TRPC6) is a voltage-independent cation channel that mediates angiotensin II-stimulated hypertrophic responses [61]. Increased intracellular Ca^2+^ plays an important role in the conversion of fibroblasts to myofibroblasts, and the production of fibrosis by activating cellular signaling [62]. TGF-β-induced upregulation of TRPC6 was inhibited by the blockade of p38 MAPK-mediated signaling [63]. TRPC6 knockout fibroblasts did not show changes in Ca^2+^ signaling, and did not promote the conversion of fibroblasts to myofibroblasts when the cells were treated with angiotensin II and TGF-β. These results demonstrate that TRPC6-Ca^2+^ signaling is essential for the induction of myofibroblasts from cardiac fibroblasts.

There are other signaling molecules involved in the induction of fibrosis. The deletion of the β-catenin gene in cardiac fibroblasts improves cardiac function and reduces fibrosis, due to the decreased production of ECM proteins by cardiac fibroblasts [64]. It has recently been found that functional primary cilia in cardiac fibroblasts are required for the canonical TGFβ signaling-induced differentiation of cardiac fibroblasts to myofibroblasts [65]. Primary cilia express polycystin-1, which is known to be a regulator of cell proliferation, cell migration, and interactions with other cells. Polycystin-1 is required for the maintenance of the cellular structures of primary cilia. When the primary cilia of cardiac fibroblasts were disrupted, specifically by deletion of the polycystin-1 gene, TGFβ-Smad3 signaling-induced ECM protein production and fibroblast differentiation were impaired. The deletion of the polycystin-1 gene enhanced pathological cardiac remodeling after myocardial infarction [66], suggesting an important role for primary cilia in hypertrophy and fibrosis. These findings also suggest that primary cilia are functional, and participate in TGFβ-induced fibrosis and myofibroblast differentiation. Heat-shock protein (Hsp) is a chaperon molecule for the conversion of fibroblasts to myofibroblasts, leading to fibrosis. Among the many Hsp proteins, Hsp47 is known to be a collagen-specific chaperone. The cardiac fibroblast-specific deletion of Hsp47 significantly reduced cardiac fibrosis and improved cardiac diastolic dysfunction after pressure overload [67]. However, the reduced collagen production in these mice increased the death rate after myocardial infarction, due to insufficient scar formation.

Collectively, these results suggest that the manipulation of the expression and activity of various signaling molecules involved in the fibrotic pathway modulates fibroblast states, leading to an alteration in fibrosis.

Angiotensin II (Ang II) is well known to be a trigger of cardiac fibrosis mediated by cardiac fibroblasts. Ang II is generated by the sequential cleavage of renin or chymase and angiotensin-converting enzyme (ACE). Ang II binds and activates angiotensin type 1 (AT1) and type 2 (AT2) receptors. ACE inhibitors and AT1 receptor antagonists are very popular drugs for the reduction in blood pressure and the treatment of heart failure. Ang I or Ang II is further metabolized to angiotensin-(1–7) (Ang (1–7)), by the action of neprilysin or ACE2 [68]. Ang (1–7) antagonizes the effects of Ang II through a G-protein-coupled receptor, the Mas receptor [69].

AT1 receptor activation induces cellular responses through G-proteins, which are linked to an increase in Ca^2+^ and the activation of Rho [69]. Ca^2+^ activates Ca^2+^-sensitive protein kinases and Ca^2+^-regulated transcription factors, leading to the induction of fibrosis. Rho activation increases the transcriptional activity of SRF to stimulate fibrotic responses.

Reactive oxygen species (ROS) have been reported to mediate cardiac hypertrophy and fibrosis. AT1 receptor stimulation generates ROS through G-protein activation [70]. ROS is a term that encompasses several highly reactive molecules, including peroxides, superoxide, and hydroxyl radicals. The ROS are thought to exert their effects mainly through cysteine modification of various proteins [71]. Although not all target molecules of ROS have been identified, ROS-mediated modification is found in protein phosphates, nuclear factor kappa B (NF-κB), and MAP. Among the various ROS, hydrogen peroxide (H_2_O_2_), generated by the AT1 receptor as well as other receptors, has been recognized as a major ROS in the redox regulation of various proteins [72]. H_2_O_2_ is more stable and membrane-permeable than superoxide or other ROS.

AT1 receptor-mediated signaling engages in crosstalk with other signaling pathways, such as the TGF-β pathway [73]. Ang II-induced cardiac hypertrophy and fibrosis was blocked in TGF-β1 knockout mice. The induction of TGF-β1 expression is mediated by AT1 receptor stimulation in vivo. However, many studies have suggested that TGF-β1 acts downstream of Ang II, and promotes fibrosis in the heart. These results demonstrate that the AT1 receptor and the TGF-β receptor interact with each other, and cardiac hypertrophy and fibrosis are regulated by a complex regulatory network of AT1 receptor- and TGF-β receptor-mediated signaling pathways, rather than by two independent signaling pathways.

AT1R signaling is also required for proper TGF-β signaling in experimental autoimmune myocarditis, by controlling Wnt/β-catenin signaling [74]. Wnt signaling is an evolutionarily conserved pathway, which plays an important role in cell-cell interactions, such as cardiomyocyte-fibroblast and endothelial cell-endothelial cell interactions. Wnt is a technical term created from the names wingless and int-1. Wnt signaling is subdivided into canonical and noncanonical signaling. β-catenin is a major player in canonical signaling [75]. β-catenin increases the transcriptional activity of transcription factor TCF (T-cell factor)/LEF (lymphoid enhancer factor), to regulate various responses, such as cell cycle and adhesion. Noncanonical signaling is grouped into the Wnt/PCP (planar cell polarity) and Wnt/Ca^2+^ pathways. Both pathways are independent of β-catenin and TCF/LEF. Wnt/PCP signaling is initiated by the binding of a noncanonical Wnt protein, such as Wnt5a or Wnt11, to one of the Fzd (frizzled) receptors and the receptor tyrosine kinase-like orphan receptor 2 (ROR2) [75]. It recruits Dvl (disheveled) to the membrane, thereby activating Rho and Rac. Activated Rho in turn activates ROCK and c-Jun-N-terminal kinase, which activates the gene expression program. The Wnt/Ca^2+^ pathway is similarly initiated by the binding of several Wnts to Fzd receptors. However, it activates G-proteins that trigger the release of Ca^2+^ from intracellular stores. Increased Ca^2+^ levels stimulate calmodulin-dependent kinase and the nuclear factor associated with T cell (NFAT), histone deacetylase 4 (HDAC4), and myocyte enhancer factor. Wnt signaling is reported to be one of the triggers of cardiac fibrosis. Although Wnt signaling is known to be activated in human cardiovascular disease [76], and the modulation of Wnt signaling has produced promising results in animal models of cardiac disease, the clinical application of Wnt signaling modulators has not yet been evaluated [77].

## 7. Control of Differentiation of Fibroblasts by Extracellular Signals and Environment

Fibroblasts exist in the interstitial spaces between cardiomyocytes, under healthy conditions. When the heart is exposed to stresses, such as myocardial infarction and hypertrophy, the fibroblasts differentiate into myofibroblasts and produce ECM components such as collagen. Fibroblasts generated at the injury site actively proliferate and form aggregates. An in vitro three-dimensional culture system that mimics several states of fibroblasts has been developed [27]. When fibroblasts are isolated and cultured in two or three dimensions, the properties of the fibroblasts change. Standard polystyrene-coated culture plates, not coated with collagen or other ECM components, were used for two-dimensional (2D) cultures, and ultra-low adhesion plates, coated with any ECM components, for the 3D cultures (Figure 1). The fibroblasts cultured on 3D structure plates formed spheres within 24 h. The morphology of the fibroblasts formed by 2D and 3D cultures was reversible, but did not depend on the tension or rigidity of the extracellular environment. A correlation of gene expression patterns was found between the 3D cultured fibroblasts and the remodeling heart being treated with isoprenaline for three weeks, or with cryo-injury treatment. However, the expression of α-SMA was decreased in the aggregates. Since the expression of α-SMA, a marker of myofibroblasts, is decreased, it cannot be said that the fibroblasts obtained in the 3D culture are conventional myofibroblasts. However, the expression pattern of the mRNA of 3D-cultured fibroblasts is similar to that of the matrifibrocytes reported by Fu et al. [78]. The analysis of fibroblasts from 3D cultures may help to elucidate the function and fate of matrifibrocytes. The 3D-cultured fibroblasts reversibly changed their morphology, and the genes expressed when they were transferred into the 2D culture. Thus, the 3D culture, but not the usual 2D culture, provides a sufficient signal to trigger remodeling. Since an in vitro system is essential for analyzing the mechanisms of differentiation, and the function of fibroblasts and myofibroblasts, the exchange of culture conditions may be a promising technique with which to analyze the complex behavior of fibroblasts.

Fibroblast fate is also regulated by the stiffness of the ECM [79]. The stiffness around the fibroblasts increases during the progression of fibrosis. Increased stiffness is sensed by integrin receptors and the actin cytoskeleton, which promotes the translocation of p38 MAPK to the nucleus and stimulates remodeling [80]. The integrin-actin cytoskeleton signaling complex also activates tyrosine kinases, such as focal adhesion kinase, Src, and Fyn. These kinases stimulate the GDP-GTP exchange of Rho, through guanine nucleotide exchange factors, leading to the activation of the Rho-ROCK-MRTF-A pathway that increases gene transcription in concerted action with SRF [81]. This pathway will be described in more detail in the YAP-TAZ signaling section.

The Hippo pathway is known to inhibit cardiomyocyte proliferation [82]. Yes-associated protein (YAP) is a transcriptional coactivator in the Hippo pathway that is negatively regulated by large tumor suppressor kinase 1 (Lats1) and Lats 2 (Figure 3).

The deletion of YAP from transcription factor 21- (Tcf21) and Col1a1-expressing fibroblasts decreased their collagen deposition, proliferation, and activation after myocardial infarction [83]. Similar decreases in angiotensin II/phenylephrine-treated fibrosis were seen when YAP was knocked down in fibroblasts [84]. Myocardin-related transcription factor A (MRTF-A) levels were decreased in YAP knockout mice, suggesting that MRTF-A expression is regulated by YAP function. These results demonstrate the importance of YAP-MRTF-A signaling for the determination of myofibroblast states, in response to ischemic and chronic stresses. Cardiomyocyte-specific deletion of YAP decreased hypertrophy, and significantly increased fibrosis [85]. Thus, YAP protects the heart against ischemic stress or pressure overload.

Lats1 and Lats2 phosphorylate YAP and inhibit YAP-mediated transcriptional activation of the Hippo pathway [86]. The deletion of Lats1 and Lats2 increased YAP activity. Cardiac fibroblast-specific deletion of Lats1 and Lats2 induced spontaneous myofibroblast differentiation [86]. Lats1 and Lats2 knockout mice showed increased fibrosis, both at baseline and after myocardial infarction. Mechanistic analysis revealed that YAP directly activates the transcriptional machinery of myofibroblasts, leading to fibrosis. These results suggest that Lats1/2-dependent YAP inhibition play an essential role in maintaining the resting state of fibroblasts.

The activities of Lats1 and Lats2 are regulated by the actin cytoskeleton, which is regulated by Rho. Rho is also involved in the MRTF-A- and MRTF-B-mediated fibrotic pathway. MRTF-A and MRTF-B help serum response factor (SRF) to bind a promotor sequence known as the serum response element (also known as the CArG box). Rho activates the transcriptional machinery, leading to fibrotic responses. Thus, Rho, Lats1/2, YAP, and MRTF-A/B form a complex network of induction of fibrotic responses.

## 8. Treatment of Fibrosis

GRK2 is a kinase that phosphorylates agonist-bound GPCRs. Using high-throughput screening, paroxetine was identified as a GRK2 inhibitor that attenuated the development of heart failure and cardiac fibrosis after myocardial infarction [87,88]. Although paroxetine is a selective serotonin reuptake inhibitor (SSRI), another SSRI, fluvoxamine, did not inhibit GRK2, and did not attenuate heart failure. These results show that a new GRK2-selective inhibitor is a therapeutic option for the treatment of heart failure and cardiac fibrosis.

In the heart, aquaporin transfers H_2_O_2_ from the extracellular space to the cytosol [89]. H_2_O_2_ entering the cells modifies proteins and changes protein functions, causing detrimental effects. Several isoforms of aquaporins are expressed in the mouse heart. Since H_2_O_2_ is generated from the superoxide anion (O_2_^−^) produced by NADPH oxidases in the extracellular space [90,91], the aquaporin-mediated transfer of H_2_O_2_ to the cytosol is critical for the induction of hypertrophy and other responses. The inhibition of H_2_O_2_ translocation, by the aquaporin 1 inhibitor bacopaside, resulted in the suppression of hypertrophic responses and fibrotic responses in vitro. The treatment of mice with a clinically approved extract containing bacopasides attenuated cardiac hypertrophy and fibrosis [92]. These results suggest that aquaporin is a promising target for the treatment of cardiac fibrosis as well as hypertrophy. They also suggest that bacopaside, or its related analogs, are safe for the treatment of cardiac fibrosis, since an extract containing bacopasides is already used as a traditional Ayurvedic medicine for memory enhancement and anti-inflammatory activity. However, aquaporins are expressed in the whole body, and the inhibition of aquaporins by synthetic drugs or antibodies may cause side effects in other tissues. It has been reported that the anti-aquaporin-4 antibody is a main cause of human neuromyelitis optica spectrum disorders, and the high level of this antibody in the plasma is linked to poor visual prognosis in humans [93].

TGF-β is a strong inducer of fibrosis. TGF-β signaling inhibitors have been used for the treatment of pulmonary fibrosis caused by bleomycin, and liver fibrosis caused by alcohol in mice. The effects of TGF-β inhibitors on cardiac fibrosis have not yet been reported in mice and humans.

Pirfenidone and nintedanib have been approved for the treatment of idiopathic pulmonary fibrosis. Pirfenidone inhibits TGF-β production, and nintedanib inhibits the tyrosine kinase activity of vascular endothelial growth factor receptor, fibroblast growth factor receptor, and platelet-derived growth factor receptor [94,95]. However, various receptor and signaling inhibitors, including these two drugs, do not exhibit direct beneficial effects on cardiac fibrosis [96]. The development of antifibrotic drugs is a difficult task. The target molecules of antifibrotic drugs in the heart and other tissues are also involved in important cellular responses such as tissue repair. Prolonged treatment with antifibrotic drugs may cause undesirable side effects.

An immunological strategy to target cardiac fibrosis has recently been reported [97]. Chimeric antigen receptor (CAR) T cells were engineered to recognize and induce ablation of myofibroblasts, leading to improved cardiac function. CAR consists of the antigen-recognizing regions of a single-chain Fv fragment, a transmembrane domain, the intracellular domains of the T-cell activation receptor CD3ζ, and the co-stimulation receptor CD28. After the binding of CAR to a myofibroblast-specific protein, mouse FAP, on myofibroblasts, CAR T cells causes cytotoxic killing, decreasing the number of myofibroblasts. CAR T cells are already used for cancer therapy, and are reported to have beneficial effects. A strategy to reduce the number of myofibroblasts is supported by the following experiments. When diphtheria toxin receptor, specifically expressed in myofibroblasts, binds diphtheria toxin, the toxin decreases the number of myofibroblasts and reduces cardiac fibrosis by myocardial infarction [98]. However, several concerns must be addressed before the application of CAR T cells to heart failure patients with serious fibrosis [99]. CAR T cells release cytokines to act on myofibroblasts, and heart failure patients are in an advanced inflammatory state. Released cytokines will therefore complicate the condition of heart failure, and may have detrimental effects on heart failure patients. Although the mouse experiments with diphtheria toxin are promising, direct extrapolation of the mouse results to human study may not be appropriate. Myofibroblasts produce ECM components such as collagen and fibronectin, which are in part protective against cardiac rupture. Another issue is the lack of information about myofibroblast-specific marker proteins in humans. A specific antigen is essential for the development of CAR T-cell-dependent treatment. Although there are several concerns, proper management of the number of myofibroblasts is expected to lead to the treatment of cardiac fibrosis, with few side effects. To reduce the side effects, a dual recognition strategy of CAR T cells may be a good option. A protein that inhibits both programmed death ligand-1 (PD-L1) and TGF-β has been successively used for the treatment of tumor growth and metastasis [100]. The dual inhibitor was more effective than treatment with the TGF-β inhibitor alone, and is expected to increase the specificity of the TGF-β inhibitor. Thus, combining a TGF-β inhibitor and an antibody that recognizes myofibroblast-specific protein into single molecule may be an exciting strategy for the treatment of cardiac fibrosis. The identification of the myofibroblast-specific antigen helps to increase the specificity of the TGF-β inhibitor against cardiac fibrosis, and restrict the action of the TGF-β inhibitor to the local area.

Table 2 summarizes some clinical trials of antifibrotic drugs. There are various drugs targeting the renin-angiotensin-aldosterone system or other signaling pathways. Since AT1 receptor signaling strongly contributes to cardiac fibrosis, the effects of AT1 receptor blockers and ACE inhibitors on cardiac fibrosis were examined. Several clinical trials showed positive outcomes. Recently, the combination therapy of AT1 receptor blocker (valsartan) and natriuretic peptide-degrading inhibitor (sacubitril) was tried, but it did not convincingly show beneficial effects on the levels of cardiac fibrosis markers compared to an ACE inhibitor (enalapril) alone [101,102].

The list does not include drugs for which the effects are evaluated using parameters associated with heart failure, such as cardiac function and the expression of marker molecules. However, the clinical significance of these trials is limited by the small number of patients. Large-scale clinical trials remain to be conducted.

β-blockers are widely used for the treatment of cardiovascular diseases including heart failure [52]. However, β-blockers are not effective for the treatment of heart failure in patients with preserved ejection that is partly caused by fibrosis [52]. Various drugs, such as relaxins, tranilast, and rosuvastatin, have beneficial effects on cardiac fibrosis in animal models, but not in human patients. The failure of many drugs to treat fibrosis in clinical trials raises several issues [118]. First, the extrapolation of results from animal models to humans is not adequate. In humans, fibrosis develops slowly, taking years to decades. In contrast, fibrosis in animal models occurs within days to months. Second, there is genetic variation between mice and humans. The genetic heterogeneity in humans may contribute to the poorer antifibrotic effects of drugs. Third, there is a difference in age between the animals used for the experiments and the patients with fibrosis. The animals are normally young, while the patients are aged people in many cases. In addition, many animal models do not mimic clinical settings. Various causes of fibrosis in humans may be another critical factor in poor clinical outcomes. Careful design of clinical trials, including the dose, timing, length of administration, and patient selection, will be necessary for beneficial outcomes.

## 9. Conclusions

Myofibroblasts are mainly differentiated from fibroblasts, and are responsible for the production of ECM at the time of injury. The relationship between fibroblasts and myofibroblasts is more complex than previously thought. Further, scRNAseq and lineage-tracing techniques have demonstrated the heterogeneity of fibroblasts, although the functional differences of various fibroblast states remain to be determined. Effective treatments for cardiac fibrosis are eagerly awaited, and the analysis of signaling in cells in various fibroblast states will help to identify therapeutic targets that are suitable for drug development. Direct reprogramming of cardiac fibroblasts, using chemical compounds to convert them to cardiomyocytes, is not presented here. Since the development of direct reprogramming technologies is currently in progress [117], these may provide another option for the treatment of cardiac fibrosis in the future.

## Figures and Tables

**Figure 1 cells-10-01716-f001:**
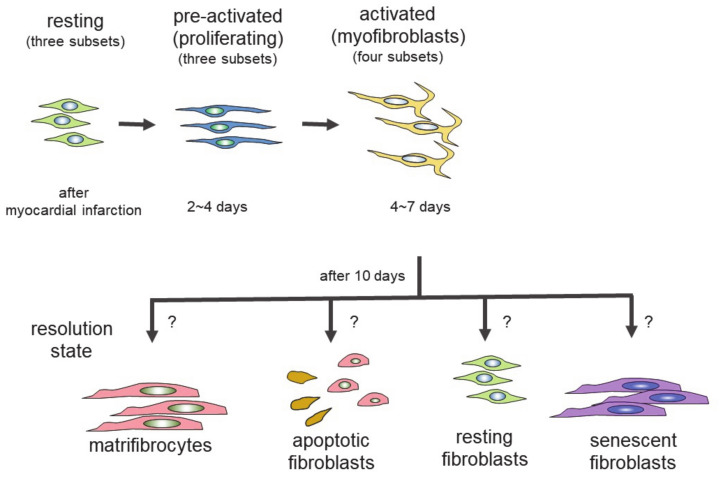
Differentiation of fibroblasts to several states of fibroblasts. Fibroblasts of each state have different proliferating activity and function. Scheme is based on two reports [33,34].

**Figure 2 cells-10-01716-f002:**
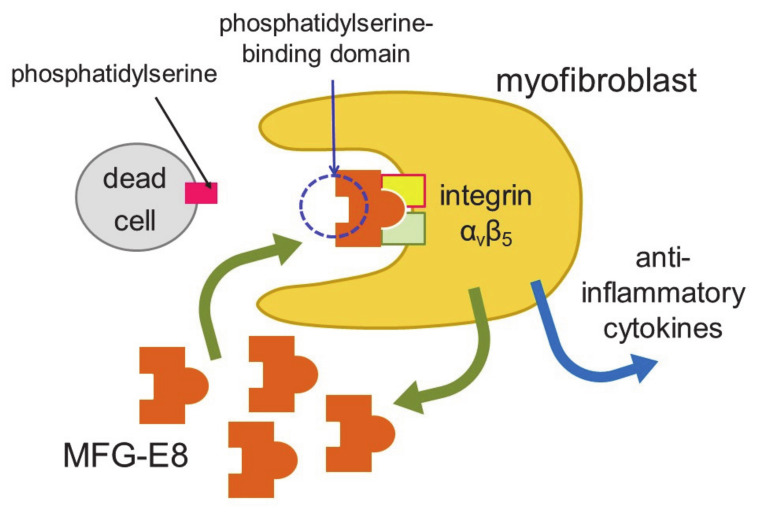
Engulfment of dead cells by myofibroblasts. Myofibroblasts engulf dead cells with the help of MFG-E8 secreted by themselves and then secrete anti-inflammatory cytokines.

**Figure 3 cells-10-01716-f003:**
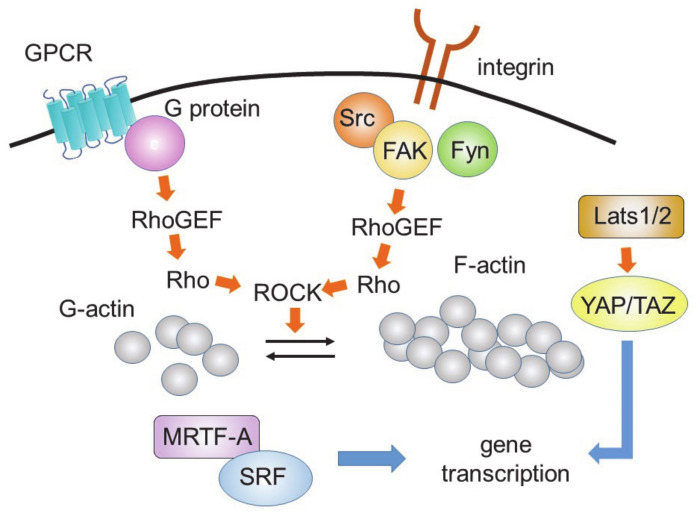
Rho- and YAP/TAZ-mediated signaling pathways.

**Table 1 cells-10-01716-t001:** Expression of marker proteins.

	States			
Marker Proteins *	Resting	Expansion (Proliferating)	Activated	Resolution
collagen type 1	+	+	+	+
DDR2	+			
PDFF receptor α	+			+
TCF21	+			+
periostin		+	+	
α-SMA			+ (but not all cells)	
CILP1, 2			+	+
COMP				+
thrombospondin 4			+	
VEGF-A			+	

Table 1 is modified from [33]. * Marker proteins: DDR2, discoidin domain receptor 2 (collagen type 1 receptor); PDGFR-α, platelet-derived growth factor receptor-α; TCF21, the transcription factor 21; α-SMA, α-smooth muscle actin; CILP, cartilage intermediate layer protein; COMP, cartilage oligomeric matrix protein; VEGF-A, vascular endothelial growth factor A.

**Table 2 cells-10-01716-t002:** Anti-fibrotic therapies of cardiac fibrosis in clinical trials.

Classification	Drug	Length of Treatment	Number of Patient (*n*)	Main Findings *
Renin-Angiotensin- Aldosterone-system inhibitors	Lisinopril	6 months	35	decrease of CVF [103]
	Losartan	6 or 12 months	19~39	decrease of CVF and PICP [104,105,106]
	Candesartan	24 months	153	suppression of cardiac fibrosis [107]
	Spironolactone	6 months	80 or 113	reduction of PIIINP [108,109]
	Eplerenone	6 or 12 months	44	decrease of PICP and PIIINP, improvement of myocardial deformation[110,111]
Statin	Atorvastatin	6 months	56	reduction of PIIINP [112]
Loop Diuretics	Torsemide	8 months	36	reduction of PICP and CVF [113]
			22	decreased PCP [114]
			24	correction of both lysyl oxidase level and increased collagen cross-linking that leads to normalization of LV chamber stiffness [115]
cGMP-specific phosphodiesterase type 5A inhibitor	Sildenafil	3 months	59	reduction of TGF-β and MCP-1 [116]

Table 2 is modified from [117]. * Abbreviation of main findings are as follows: CVF, collagen volume fraction; PICP, the carboxy-terminal peptide of procollagen type I; PIIINP, the amino-terminal peptide of type III procollagen; PCP, procollagen type I carboxy-terminal proteinase; LV, left ventricular; MCP-1, monocyte chemoattractant protein-1.

## Data Availability

This review does not contain original data.

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
