# Peer review of "Cardiac Fibrosis and Fibroblasts"

_cells, 2021, doi:10.3390/cells10071716_

Round 1

Reviewer 1 Report

The author provides a review of myofibroblasts and fibrosis in the heart.  The article is generally well-written, although grammatical editing is needed.  Several relatively minor suggestions below that may enhance the clarity and thoroughness of the review:

- In the first sentence of Section 1, expand to include other cell types in the heart.

- In Section 1, the author indicates that the “myofibroblasts are the only cells that produce extracellular matrix”.  Suggest changing this to “primary cells” since there are other cell types that produce extracellular matrix.

- In Figure 1, the bottom portion indicating 2D cultured fibroblasts and 3D cultured fibroblasts is a bit confusing as this is not discussed in the text until several pages later.  Suggest removing that part of the figure.

- In Section 1, the author discusses research illustrating four states of fibroblasts in a myocardial infarction model.  Has this research been expanded to other disease models or is it unique to myocardial infarction?

- In the last paragraph of Section 1, the author speculates on a role for ITIH1 in myofibroblast or matrifibrocyte formation or fibrosis.  As written, this appears to be largely speculation.  This should be removed or the discussion expanded to provide additional evidence for the role of ITIH1.

- Section 3 is titled “Differentiation of fibroblasts to other cells”; however, the discussion in this section is a bit confusing as the first paragraph and part of the second paragraph focuses on the conversion of other cells to fibroblasts or myofibroblasts and not the conversion of fibroblasts to other cell types.

- The author indicates in Section 4 that myofibroblasts secrete cytokines that inhibit the inflammatory process.  It would be beneficial to expand the discussion of the contribution of myofibroblasts to the inflammatory process as there is also literature suggesting an inflammatory role for these cells.

- In Section 8, the author discusses treatment of fibrosis and heart failure.  It would be beneficial to focus this discussion on fibrosis since this is the topic of the review.  It is not clear how some of the heart treatment options are relevant and this detracts from the focus.  It would be helpful to discuss briefly why advancements in treating fibrosis have been slow to develop.

Reviewer 2 Report

The review article by H. Kurose aims to summarize current knowledge about cardiac myofibroblasts and fibrosis. This is a timely and interesting piece of work with some non-obvious, but exciting paragraphs like the one on immunotherapy. On the other hand, some parts seem to be incomplete lacking important informations. Furthermore, the structure and content of certain paragraphs could be improved, as in present form it might be confusing for the reader.

Generally, the author does not focus exclusively on myofibroblasts, but extends the review on cardiac fibroblasts. It is logical and understandable, as myofibroblasts represent a subset of activated fibroblasts. However, in its present form, the content does not reflect a title (the author may consider to change it).

Specific comments:

  • overall cardiac fibrosis and myofibroblasts in inflammatory heart disease are not mentioned, while cardiac and systemic inflammation represent important triggers of cardiac fibrosis
  • where applicable, description of human data and clinical studies should be provided
  • listing of myofibroblast markers would greatly help the readers to properly interpret original data, as most of advanced mouse data refer to cardiac fibroblasts (but not myofibroblasts) - this should be clearly stated
  • Contractility is an important feature of myofibroblasts and this is not mentioned in the manuscript
  • Paragraph 3 should describe differentiation of fibroblasts to other cells, whereas half of this paragraph describes possible origin of cardiac fibroblasts. The structure should be improved (perhaps in two paragraphs). Inflammatory CD133+ cells should be also mentioned as a possible source of myofibroblast-like cells in inflammatory heart disease.
  • Wnt and angiotensin II represent important triggers for cardiac fibrosis and for myofibroblasts, which are not presented in the paragraph 5. Currently only b-catenin is briefly mentioned. Please add.
  • It should be justified and clarified to the readers why Yap/Taz pathway is described in a separate paragraph (logically it belongs to paragraph 5).
  • Targets for treatment of fibrosis described in paragraph 8 should refer to profibrotic signalling pathways described in paragraph 5.
  • As this review is focused on the heart, the relevant studies for treatment of cardiac fibrosis should be mentioned in paragraph 8. In addition to highly specialized potential treatment options, the conventional blockade of key profibrotic pathways like TGFb, AngII, Wnt/b-catenin etc should be mentioned as well. Examples from SSc and lung fibrosis can be highlighted, but a specific justification should be provided. Otherwise non-cardiac studies are confusing

Minor comments:

  • the first sentence of the abstract is unclear
  • paragraph 1, first sentence: please add endothelial cells
  • paragraph 1: the sentence "Myofibroblasts are only the cells that produce extracellular matrix" is confusing
  • Paragraph 8: does Bacopaside affect fibrosis?

Reviewer 3 Report

This is overall a well-written review on the functional aspect of myofibroblast in cardiac fibrosis.

However, several grammar/spelling errors need to be corrected. 

  1. section 1, second paragraph, line 3 "since maker proteins" should be "marker proteins". the third paragraph, line 1 "about" should be deleted. the fifth paragraph, line 7 "with other studies" should include refereces.
  2.  in the last paragraph of section 1, the author discussed ITIH1 which has nothing to do with the rest of this paragraph. This is confusing to readers. 
  3. section 5, second paragraph, line 3, "cardiac fibroblasts ae blocked by ..." 

Round 2

Reviewer 2 Report

The manuscript is substantially improved and reads well now. Few minor comments:

  • ref 20 is not referring to cardiac myofibroblasts. Please exchange citation or rephrase.
  • line 405-13: AngII signaling is required for proper TGFb signaling (incl. activation of b-catenin pathway) PMID: 33576779
  • Section 7: ATR and ACE blockers should be mentioned
  • Section 7: TGFb inhibitors are not used because of toxicity

Author Response

Comments

The manuscript is substantially improved and reads well now. Few minor comments:

ref 20 is not referring to cardiac myofibroblasts. Please exchange citation or rephrase.

Thank you for the comment. I changed the sentence and reference. Line numbers are 87 to 88.

line 405-13: AngII signaling is required for proper TGFb signaling (incl. activation of b-catenin pathway) PMID: 33576779.

Thank you for the comment. I added the sentence that AT1R signaling is required for TGF-β signaling with reference (reference 73). Line numbers are 414 to 415.

Section 7: ATR and ACE blockers should be mentioned.

Thank you for the comment. I added the sentences about AT1R blocker and ACE inhibitor. Line numbers are 581 to 584.

I also added the combination therapy of AT1R blockade and natriuretic peptide inhibition. Line numbers are 584 to 587.

Section 7: TGFb inhibitors are not used because of toxicity.

Thank you for comment. I forgot to mention that the effects of TGF-β inhibitors are examined in animal models of fibrosis. I added the word ‘in mice’. Line number is 536.

I also changed the sentence ‘The effects of TGF-β inhibitors on cardiac fibrosis have not yet been examined’ to ‘The effects of TGF-β inhibitors on cardiac fibrosis have not yet been examined in mice and human’. Line numbers are 536 to 537.